

# Early childhood caries and its associations with sugar consumption, overweight and exclusive breastfeeding in low, middle and high-income countries: an ecological study

Morenike O. Folayan[1,*], Maha El Tantawi[2,*], Francisco Ramos-Gomez[3] and Wael Sabbah[4]

[1] Department of Child Dental Health, Obafemi Awolowo University, Ile-Ife, Nigeria
[2] Department of Pediatric Dentistry and Dental Public Health, Faculty of Dentistry, Alexandria University, Alexandria, Egypt
[3] Division of Growth & Development, Section of Pediatric Dentistry, University of California, Los Angeles, Los Angeles, CA, USA
[4] Faculty of Dentistry, Oral & Craniofacial Sciences, King's College London, University of London, London, UK
* These authors contributed equally to this work.

Corresponding author
Morenike O. Folayan,
toyinukpong@yahoo.co.uk

## ABSTRACT

**Aim:** This ecological study examined the associations between the prevalence of early childhood caries (ECC), overweight, country's per capita sugar consumption and duration of exclusive breastfeeding.

**Methods:** Per capita consumption of sugar in kilograms, percentage of children exclusively breastfed until 6 months of age, percentage of 0–5-year-old children with overweight status, and percentage of 3–5-year-old children with ECC were compared among low-income countries (LICs), middle-income countries (MICs) and high-income countries (HICs). The association between the prevalence of ECC and the study variables, and the effect modification by income region were assessed using multivariable linear regression models. Regression coefficients, confidence intervals, partial eta squared and $P$-values for effect modification were calculated.

**Results:** The per capita sugar consumption in LICs was significantly lower than in MICs ($P = 0.001$) and HICs ($P < 0.001$). The percentage of infants who exclusively breastfed up to 6 months was significantly lower in HICs than in LICs ($P < 0.001$) and MICs ($P = 0.003$). The prevalence of overweight was significantly lower in LICs than in MICs ($P < 0.001$) and HICs ($P = 0.021$). The prevalence of ECC was significantly lower in HICs than in MICs ($P < 0.001$). Income was a significant modifier of the associations between the prevalence of ECC, per capita sugar consumption ($P = 0.005$), and exclusive breastfeeding up to 6 months ($P = 0.03$). The associations between the prevalence of ECC and per capita sugar consumption at the global level and for MICs were stronger (partial eta squared = 0.05 and 0.13 respectively) than for LICs and HICs (partial eta squared <0.0001 and 0.003 respectively). Only in MICs was there a significant association between the prevalence of ECC and per capita sugar consumption ($P = 0.002$), and between the prevalence of ECC and the percentage of children exclusively breastfed up to 6 months ($P = 0.02$).

**Conclusion:** Though the quantity of sugar consumption and exclusive breastfeeding may be a significant risk indicator for ECC in MICs, sugar consumption may be more of a risk indicator for ECC in HICs than in LICs, and vice versa for exclusive breastfeeding. Although ECC and overweight are both sugar-related diseases, we found no significant relationship between them.

# INTRODUCTION

Caries and obesity have common risk factors—diet and nutrition (*Cregger et al., 2017*). The dietary factor that links the risk factors is high sugar consumption at the expense of vegetables and fruits (*Alswat et al., 2016*). Greater access to sugar and refined carbohydrates increases the risk for caries and obesity. For example, during war times in Japan, access to sugar was highly restricted, and caries prevalence dropped significantly, only to increase after the war when access to sugar increased (*Takahashi, 1959*; *Takeuchi, 1961*). Sugar taxation also has reduced sugar consumption and lowered the risk of dental caries and obesity (*Yoshida & Simoes, 2018*; *Schwendicke et al., 2016*), whereas individuals who live in proximity to grocery stores and food stores have greater access to sugar with higher risk for caries and obesity (*Baker et al., 2006*; *Tellez et al., 2006*; *Jetter & Cassady, 2006*; *Turrell et al., 2009*; *Bodor et al., 2010*; *Broomhead, 2017*). Although these studies were mostly conducted in Western, high-income countries, similar associations have been observed in several low- and middle-income countries. For example, nutrition transition (increase intake of refined carbohydrates) associated with higher income in low- and middle-income countries is also associated with higher prevalence of obesity and caries (*Enwonwu, 2010*; *Gracey & King, 2009*; *Popkin, Adair & Ng, 2012*).

Breastfeeding modulates the risk of caries and obesity. Exclusive breastfeeding for the first 6 months is protective against overweight and obesity in childhood and beyond (*Arenz et al., 2004*; *Harder et al., 2005*; *Plagemann & Harder, 2005*; *Horta & Victora, 2013*; *Weng et al., 2012*). Multiple pathways have been suggested for this relationship, one of which is exclusive breastfeeding precluding inappropriate feeding practices, such as early introduction of complementary foods that are high in free sugar (*World Health Organization, 2014*). On the other hand, breastfeeding more than three times a day (*Feldens et al., 2018*) and beyond 24 months may increase the risk for early childhood caries (ECC) (*Peres et al., 2017*). To our knowledge, no study has addressed the relationships between caries, obesity and exclusive breastfeeding.

Numerous micro-level studies have been conducted on the link between sugar consumption, obesity and ECC, but none has been conducted to identify the relationship between these factors at the macro-level. Macro-level studies on the relationships between per capita sugar consumption, obesity and ECC may give insights about structural interventions that could help achieve population-level disease control. Given the lack of studies in this area and the potential benefit of such information for health promotion

policies, we set out to determine the relationship between sugar consumption and the prevalence of ECC, exclusive breast feeding, and overweight status at the global and regional levels. This ecological study is a hypothesis-generating study on the relationships between the prevalence of ECC, overweight (which is an indicator of high sugar consumption), exposure to breast milk, and country per capita sugar consumption in three World Bank defined income regions.

## MATERIALS AND METHODS

An ecological design was used to determine the relationship between ECC and study variables (per capita sugar consumption, exclusive breastfeeding in the first 6 months of life and overweight) at the global and income region levels. All data were obtained from open sources as explained below.

### Percentage of 3–5-year-old children with ECC

Early childhood caries was defined as the presence of one or more decayed, missing due to decay or filled primary tooth surfaces in children less than 72 months of age (*Drury et al., 1999*). The data on the prevalence of ECC in 3–5-year-old children extracted from the World Health Organization (WHO) country oral health profile database and other online databases published between 2007 and 2017 by *El Tantawi et al. (2018)* were used for this analysis. No language filter was used for data extraction. The retrieved data were used to calculate the percentage of children with ECC for each of the 193 countries listed by the UN for which data was available. The prevalence of ECC for 3–5-year-old children was computed by dividing the number of children affected by ECC by the total number of children examined in the country, multiplied by 100.

### Per capita sugar consumption

Data on sugar consumption were collected from the *US Department of Agriculture (2019)*, *Euromonitor International (2019)* and Statista *Protectivity Insurance (2018)*. Consumption was measured in per capita kilogram per country for 2017. These data are national averages for all age groups; no age-specific data were available.

### Measure of overweight

Information on nutritional status was obtained from country-level data produced by *United Nations Children's Fund, World Health Organization, World Bank Group (2018)* and *UNICEF (2018)*, covering the period 2007 to 2017. The threshold for defining overweight was established by the WHO-UNICEF Technical Advisory Group on Nutrition Monitoring in relation to standard deviations of the normative WHO Child Growth Standards. Overweight status in children aged 0–5 years old was defined as one standard deviation above median weight-for-height. The prevalence of overweight was calculated as the percentage of children 0–5-year-old children who met the definition.

### Exclusive breastfeeding

Country-level estimates of the percentage of infants exclusively breastfed for the first 6 months of life were obtained from the WHO Global Health Observatory data repository

(*World Health Organization, 2018*). We used the latest available estimates for the period 2007–2017.

## Economic level

The economic level of countries was based on the 2017 Gross National Income per capita (GNI), calculated with the World Bank Atlas method (2018). This level was used to group countries into three income categories—low, middle and high. The middle-income category was derived by combining the lower middle and upper middle-income categories. The income regions were as follows: low-income (LICs–GNI of US$1,025 or less); middle-income (MICs–GNI of US$1,026–$12,475); and high-income (HICs–GNI of US $12,476 or more) (*The World Bank Group, 2018*).

## Data analysis

Per capita consumption of sugar in kilograms (kg), percentage of children exclusively breastfed until 6 months of age, percentage of children 0–5 years old with overweight status, and percentage of 3–5-year-old children with ECC were assessed for normality with Kolmogorov–Smirnov and Shapiro–Wilks tests, QQ plots, and histograms. The percentage of 0–5-year-old children with overweight status and sugar consumption per capita were skewed. Normally distributed variables were compared among the three income regions using ANOVA followed by Scheffé's test for post hoc pairwise comparisons. The Kruskal–Wallis test was used to compare the percentage of 0–5-year-old children with overweight status and per capita sugar consumption in kg among income regions followed by post hoc pairwise comparisons using the Dunn–Bonferroni test to adjust for multiple testing.

Multivariable linear regression analysis was used to assess each of the following relationships in three separate models: associations between the prevalence of ECC and per capita sugar consumption, exclusive breastfeeding and overweight status adjusted for income region. Three other models were constructed to assess the interaction (effect modification) of income region on the associations between ECC prevalence and the other study variables. Regression diagnostics were checked to verify models' assumptions. These included drawing scatter diagrams of ECC prevalence against the explanatory variables to check the linearity of the relationships, P-P plots and histograms to assess normality of residuals, and scatter plots of residuals against predicted values to check that the variance of residuals were constant (Appendix A, Figs. 1–7). Regression coefficient, 95% confidence intervals and partial eta squared, as a measure of effect size were calculated. SPSS version 22 (IBM Corp., Armonk, NY, USA) was used for statistical analysis. Two-sided significance was set at 5%.

## RESULTS

Combined data for the prevalence of ECC and per capita sugar consumption per income region were available for 77 countries (4 LICs, 43 MICs and 30 HICs), and ECC and percentage of children exclusively breastfed were available for 57 countries (6 LICs, 37 MICs and 14 HICs). Also, ECC and the percentage of 0–5-year-old children with

**Table 1 Per capita sugar consumption, exclusive breastfeeding up to 6 months, overweight and ECC in LICs, MICs and HICs.**

| Variables | LICs | MICs | HICs | P value |
|---|---|---|---|---|
| Per capita sugar consumption in kilograms: Median (IQR) | 9.24 (8.22)[a] | 31.94 (21.13)[b] | 30.48 (17.80)[b] | 0.007* |
| Percentage of infants with exclusive breastfeeding up to 6 months: Mean (SD) | 53.37 (11.72)[a] | 40.01 (16.75)[a] | 15.41 (10.83)[b] | <0.001* |
| Percentage of 0–5 year old overweight children: Median (IQR) | 3.25 (3.40)[a] | 7.05 (7.37)[b] | 7.70 (1.4)[b] | 0.03*,¶ |
| Percentage of 3- to 5-year-old children with ECC: Mean (SD) | 63.12 (20.33)[a,b] | 65.65 (17.85)[a] | 46.01 (23.33)* | <0.001* |

Notes:
* Statistically significant at $P < 0.05$.
[a] Letter denote statistically significant differences.
[b] Letter denote statistically significant differences.
¶ The Kruskal–Wallis test was used followed by the Dunn–Bonferroni test to adjust for multiple pairwise comparisons. For the other variables, ANOVA was used followed by Scheffé's test to adjust for multiple pairwise comparisons.
LICs, low-income countries; MICs, middle-income countries; HICs, high-income countries; IQR, inter quartile range.

overweight status were available for 53 countries (6 LICs, 40 MICs and 7 HICs), and ECC and income region data were available for 85 countries (6 LICs, 45 MICs and 34 HICs). Further analysis was restricted per variable for these numbers. The additional (SPSS) file (Appendix B) includes information on all the variables extracted for this study.

Table 1 highlights the associations between sugar consumption, breastfeeding, overweight status and ECC and the three income regions. Per capita sugar consumption was significantly lower in LICs than in MICs (median = 9.24 kg vs 31.94 kg, $P = 0.01$) and HICs (median = 9.24 kg vs 30.48 kg, $P = 0.005$). The percentage of infants who were exclusively breastfed up to 6 months was significantly lower in HICs when compared to LICs (mean = 15.41 vs 53.37, $P < 0.001$) and MICs (mean = 53.37 vs 40.01, $P < 0.001$). The prevalence of overweight status in 0–5-year-old children was significantly lower in LICs when compared to MICs (median = 3.25 vs 7.05, $P = 0.007$) and HICs (median = 3.25 vs 7.70, $P = 0.039$). The percentage of 3–5-year-old children with ECC was significantly lower in HICs when compared to MICs (mean = 46.01 vs 65.65, $P < 0.001$). No significant differences were observed in this variable in LICs and MICs (mean = 63.12 vs 65.65, $P = 0.960$) or that between LICs and HICs (mean = 63.12 vs 46.01, $P = 0.172$).

Table 2 shows the association between the percentage of 3–5-year-old children with ECC and the other study variables. Income regions significantly modified the association between the prevalence of ECC and per capita sugar consumption ($P$ for interaction = 0.005) as well as the prevalence of ECC and the percentage of children exclusively breastfed ($P$ for interaction = 0.03). The only significant associations were the positive association between the prevalence of ECC and per capita sugar consumption in MICs ($P = 0.002$), and the positive association between the prevalence of ECC and the percentage of children exclusively breastfed in MICs ($P = 0.02$).

There was a positive association between the prevalence of ECC and per capita sugar consumption at the global level (regression coefficient = 0.18), in MICs (regression coefficient = 0.52) and in HICs (regression coefficient = 0.05) such that countries with high per capita sugar consumption had high prevalence of ECC. On the other hand, the association between the prevalence of ECC and per capita sugar consumption in LICs (regression coefficient = −0.004) was inverse such that countries with high per capita sugar consumption had low ECC prevalence. The associations between the prevalence of

**Table 2** Association between the percentage of 3–5 year old children with ECC and per capita sugar consumption, percentage of infants with exclusive breastfeeding up to 6 months and percentage of 0–5 years of age children who are overweight by income region and their effect modification (with complete dataset, including UAE per capita sugar consumption).

| Variables | Association with the percentage of 3–5 year old children with ECC regression coefficient (95% CI), (partial eta squared) | | | | |
| --- | --- | --- | --- | --- | --- |
| | LICs | MICs | HICs | Global[¶] | P value for interaction |
| Per capita sugar consumption | −0.004 [−2.50 to 2.50], (<0.0001) | 0.52 [0.20–0.83]*, (0.13) | 0.05 [−0.15 to 0.24], (0.003) | 0.18 [−0.01 to 0.37], (0.05) | 0.005* |
| Exclusive breastfeeding | 0.26 [−0.14 to 0.66], (0.03) | 0.39 [0.07–0.72]*, (0.10) | −0.004 [−0.85 to 0.84], (0.000002) | 0.11 [−0.24 to 0.47], (0.008) | 0.03* |
| Overweight in under 0–5 year old children | −0.76 [−5.79 to 4.27], (0.002) | 0.33 [−0.64, 1.31], (0.01) | −1.60 [−3.96 to 0.75], (0.04) | 0.30 [−0.69 to 1.29], (0.008) | 0.26 |

**Notes:**
¶ Adjusted for income region.
* Statistically significant at $P < 0.05$.
LICs, low-income countries; MICs, middle-income countries; HICs, high-income countries; CI, confidence interval.

ECC and per capita sugar consumption at the global level and for MICs were stronger (partial eta squared = 0.05 and 0.13 respectively) than that for LICs and HICs (partial eta squared <0.0001 and 0.003 respectively).

There was a direct association between the prevalence of ECC and the percentage of infants who were exclusively breastfed up to 6 months at the global level (regression coefficient = 0.11), in LICs (regression coefficient = 0.26) and MICs (regression coefficient = 0.39) although this was significant only in MICs ($P = 0.02$). The association between the prevalence of ECC and the percentage of infants who were exclusively breastfed up to 6 months in HICs was inverse (regression coefficient = −0.004). The associations of LICs and MICs (partial eta squared = 0.03 and 0.10) were stronger than that observed at the global level (partial eta squared = 0.008) and in HICs (partial eta squared = 0.000002).

There was no statistically significant modification by income region for the association between the prevalence of ECC and the prevalence of overweight ($P$ of interaction = 0.26).

## DISCUSSION

To our knowledge, this is the first ecological study to determine the association between per capita sugar consumption, the prevalence of exclusive breastfeeding, overweight status and ECC in various income regions. The results highlight a complex relationship between these factors and the prevalence of sugar-related diseases (ECC and overweight). Three especially important study outcomes were these: The association between per capita sugar consumption and ECC varied significantly by income regions, and HICs had the lowest prevalence of ECC despite having the highest quantity of sugar consumption. Also, although LICs had low sugar consumption, their prevalence of ECC was not significantly different from that of MICs that had higher sugar consumption. A per capita sugar consumption was positively associated with the prevalence of ECC in MICs and HICs, though the association was only significant with MICs. This direct association meant that in these countries with high sugar consumption per capita, the prevalence of ECC was

also high. Second, exclusive breastfeeding for 6 months was associated with higher prevalence of ECC in LICs and MICs, though this association was only significant for MICs. The association in HICs was inverse, weak and non-significant statistically. Third, there was no significant association between ECC and overweight and no significant effect modification by income region.

We found no relationship between ECC and overweight status even though both diseases are sugar related. Future studies are needed to clarify the relationships between these diseases before common country-level interventions for ECC and obesity are designed based on the assumption that both diseases have a common disease pathway and that the impact of sugar control may be similar for both diseases.

Direct association between ECC and exclusive breastfeeding observed at the global level was present in LICs and MICs but not in HICs although only the association in MICs was significant. This finding may be attributed to confounding by wealth or other factors, which emphasizes that the possible association between ECC and breastfeeding warrants further investigation. At the micro level, prolonged breastfeeding beyond 12 months had been indicted as a risk factor for ECC (*Victora et al., 2016*). However, country-level data on duration of breastfeeding for 12 months or more were not available to us, so it was not possible to assess the association between duration of breastfeeding and ECC. Our study suggests that exclusive breastfeeding for as little as 6 months may be a risk indicator ECC in some countries but not others. The differences among countries in this association by income region may be attributed to the confounding effect of wealth or oral hygiene. *Victora et al. (2016)* suggested that good oral hygiene may mediate the association between ECC and breastfeeding at the individual level. There are no data on country-level oral hygiene status by age; therefore, it was not possible to determine the mediating effect of oral hygiene on the associations that we observed.

In addition, our observation that the prevalence of ECC was high in LICs despite the low consumption of sugar may be explained by the impact on ECC of frequency of sugar consumption compared with the quantity of sugar consumed. Studies have shown that both the quantity (*Skafida & Chambers, 2018*; *Van Loveren, 2019*) and frequency of sugar consumption (*Van Loveren, 2019*) are important in the cause of ECC, but the frequency of sugar consumption may be more of a risk factor than is the quantity of sugar consumed (*Van Loveren, 2019*). There are no global data on the frequency of sugar consumption, so we could not include that variable in the analysis.

Our observation that per capita sugar consumption and overweight status were higher in MICs than in LICs may be explained by increasing country affluence, which is associated with greater access to processed than to agriculture-based diets (*Lagerweij & Van Loveren, 2015*; *Mascarenhas, 2016*). Dietary transition probably occurs faster than do changes that can support access to dental care that reduces the risk of sugar-related diseases. The low prevalence of ECC in HIC despite high per capita consumption of sugar and the significantly lower prevalence of ECC in HICs compared to MICs support this proposition.

The association between per capita sugar consumption and the prevalence of ECC was positive globally, in MICs and HICs but not in LICs. Perhaps there is a dose response

association between sugar consumption and ECC (*Van Loveren, 2019*) beyond which other factors have stronger impact, such as access to health care or healthful foods such as fruits, vegetables and milk (*Moynihan & Petersen, 2004*; *MacKeown, Cleaton-Jones & Edwards, 2000*; *Mayén et al., 2014*). This possibility should be studied further and supports the call to avoid generalization of results obtained from studies in one setting to another.

Control of sugar intake in preschool children is important because dietary habits established in early childhood likely are maintained throughout life. Indeed, research suggests a correlation between excessive sugar intake and suicide ideation in adolescents (*Pan, Zhang & Shi, 2011*; *Jacob, Stubbs & Koyanagi, 2019*), dementia (*Stephan et al., 2010*) and cancer mortality (*Malik et al., 2019*). Even continuous intake of sugar at low quantity seems to increase the risk for caries in later life (*Sheiham & James, 2015*). However, while individual actions to reduce sugar consumption may affect the risk for ECC, our study result suggests that country-level actions may have macro-level impact in MICs and HICs but not in LICs. This finding does not preclude the need for reduced sugar consumption as an individual-level effort to reduce the risk for ECC. These observations need further investigation.

Our results must be interpreted with caution. Ecological fallacy is associated with this type of study, and it is greater at the macro-level than it is at the micro level. This difference is due to aggregation of data resulting in the loss or concealment of certain information (*Saunders & Abel, 2014*; *Hsieh, 2019*). Another limitation of our study is its cross-sectional design. It is important to acknowledge that the findings do not prove causality. Also, due to limited country-level data, we were unable to adjust for traditional ECC risk factors, such as oral-hygiene, access to dental care and fluoridated water, and the use of fluoridated toothpaste at a young age. Water fluoridation may be particularly important for causing differences in ECC by income region as well as inequalities in the association between risk factors and ECC prevalence across regions. Future research should address this important point.

We also were limited in our analysis on per capita consumption of sugar and prevalence of ECC because disaggregated data on sugar consumption by age, sex and socioeconomic status were not accessible. Disaggregated data are urgently needed in view of the growing recognition of the role of sugar in several non-communicable diseases. For example, the Institute of Health Metrics and Evaluation Global Burden of Disease research provides age- and sex-related estimates for tobacco smoking, chewing tobacco and secondhand smoking as comprehensive risk factors for tobacco diseases. However, the Institute provides estimates only for diet high in sugar-sweetened beverages, which limits the ability to quantify the prevalence, incidence, burden and mortality attributed to consumption of all dietary forms of sugar intake (*Institute of Health Metrics, 2017*).

Despite these limitations, the study assessed effect modification by income region that could be considered in the planning of future regional and/or country specific studies. Also, our findings add new information on the macro-level relationship between sugar consumption and ECC, and the relationship between ECC and overweight in pre-school

children. The findings highlight differences in sugar consumption-associated risk for ECC by income regions. The study by *El Tantawi et al. (2018)* showed that HICs may have better structural factors—universal health coverage and growth of GNI—that mitigate the risk of ECC. Therefore, despite high sugar consumption in HICs, access to dental care may be a risk-limiting factor for ECC.

## CONCLUSIONS

In conclusion, although ECC and overweight have a common risk factor in terms of sugar consumption, the prevalence of ECC was not significantly associated with the prevalence of overweight. The quantity of sugar consumed and prevalence of exclusive breastfeeding were not significantly associated with ECC prevalence globally and were related differently in various income regions. The findings of this study suggest the need to assess diversities in the association between ECC and sugar consumption in countries and individuals with various income levels and the effect of breastfeeding on ECC prevalence in various income regions.

### Funding
The authors received no funding for this work.

### Competing Interests
Morenike Oluwatoyin Folayan and Maha El Tantawi are Academic Editors for PeerJ. The other authors declare no competing interest.

### Author Contributions
- Morenike O. Folayan conceived and designed the experiments, performed the experiments, prepared figures and/or tables, authored or reviewed drafts of the paper, and approved the final draft.
- Maha El Tantawi performed the experiments, analyzed the data, prepared figures and/or tables, authored or reviewed drafts of the paper, and approved the final draft.
- Francisco Ramos-Gomez performed the experiments, authored or reviewed drafts of the paper, and approved the final draft.
- Wael Sabbah performed the experiments, authored or reviewed drafts of the paper, and approved the final draft.

### Data Availability
   The raw data are available in the Supplemental Files.

### Supplemental Information
Supplemental information for this article can be found online at http://dx.doi.org/10.7717/peerj.9413#supplemental-information.

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
