# Peer review of "Early childhood caries and its associations with sugar consumption, overweight and exclusive breastfeeding in low, middle and high-income countries: an ecological study"

_PeerJ, doi:10.7717/peerj.9413_

## Round 0.1 · original submission · Major Revisions

Ecological studies are always challenging and the reviewers have asked some important questions that you will need to address in a revised version of your manuscript, along with responding to my comments further below. Each comment should be individually responded to with either a justification for not making any change(s) in response or a description of the change related to that comment.

As Reviewer #1 alludes to, parametric tests such as ANOVAs make assumptions about the distribution of the model residuals. You either need to argue that these assumptions are satisfied here (describing the model diagnostics in your methods, note that this is likely to be challenging given the sample sizes) or use another approach (such as Kruskal-Wallis) that does not make unjustifiable assumptions. While the data is from publicly available sources, these are not always clear (e.g. Lines 119–120) and it would be better to have the data included alongside the manuscript so readers can replicate the analyses and consider alternative approaches. Unless there are specific reasons why this cannot be done, please provide the data with your revised manuscript. Reviewer #1 also raises an important question about fluoride use (including water fluoridation). Can you add this component to the analyses? Their other comments all warrant careful thought and responses.

Reviewer #2 also asks for clearer methods, which will include the statistical methods along with data acquisition, and more consideration of fluoride. You seem to be suggesting on Lines 250–252 that fluoride toothpaste use is not available, but what about fluoridation of water supplies, which this reviewer also asks about? While this absolutely needs to be included in the discussion, I’d prefer to see it also incorporated into the analyses if at all possible. They also raise the question of datasets, including those referred to on Lines 119–120. Their other comments will also be useful to you and all warrant considered responses in your rebuttal.

To the reviewer’s comments, I will add the below. Please note also that there are some typos sprinkled throughout the manuscript and careful proofreading should be performed on the revised version prior to resubmission.

1) It’s not clear why ECC prevalences are categorised in the first place as this always results in a loss of information. Why not look at Spearman’s correlations instead when looking at ECCs alongside another continuous measure (to avoid assuming linearity with Pearson’s correlations) and show the data through scatterplots? If you feel that the current approach would be preferably to testing for monotonicity, please do explain your argument, but otherwise, this more general test would simplify your presentation and reporting as well as increasing your power compared to other non-parametric approaches.

2) It should always be clear to the reader exactly what numbers refer to. For example, Line 42 does not make it clear what these two values are, although a good guess can certainly be made.

3) As well as being careful with the analyses, it is also important to be careful about the summary statistics. For example, the sugar consumptions (Line 44) and percentages overweight (Lines 44–45) are clearly skewed. Several values in Table 1 are also appreciably skewed and/or contain outliers. You might prefer to report medians and IQRs (or 25th and 75th percentiles) if you change to making comparisons with Kruskal-Wallis.

4) While p-values are not the be all and end all, when you say that values are higher/highest/lowest/etc., please provide these so it is clear when you mean that the differences are numerical and when they are statistical. Better is to provide effect sizes and if you use correlations, these can usefully accompany p-values in Lines 46–52.

5) If you wish to claim different relationships by income region (Lines 55–56), you will need to statistically test the evidence for this effect modification (interaction). See also Lines 221–222, 282–283, 313–314, etc. This would be easier with a linear model such as ANOVA but could be done for other approaches, although you might need to bootstrap to perform these tests.

6) The abstract’s conclusions would be stronger with some implications added here.

7) Remember to give currencies when appropriate (Lines 147–148).

8) Please indicate the sided-ness of significance on Line 165.

9) What is “[t]he additional file 1 includes information on the countries per economic block and information on all the variables extracted for this study”?

10) Given the number of comparisons being performed here, please be careful to note a lack of statistical significance early in the sentence, e.g. Lines 176–177.

11) Please be consistent in your decimal places for p-values (e.g. “P<0.0001” on Line 178, “P=0.001” and “P=0.01” on Line 179). I suggest 3 decimal places in all cases with “P<0.001” for smaller p-values.

12) Note that describing statistical significance (Line 177 onwards) is not useful in itself as the magnitude of the difference or association is also important. Please add measures of effect and, where available, uncertainty (e.g., confidence intervals). In particular, “P=0.20” and “P=0.51” (Line 195) are not providing evidence of any trends (see elsewhere also) and you need to be very clear in explaining when there is no evidence and do so early in the sentence or paragraph to help the reader follow the findings. While you might mean numerically “lower” on Line 206, these results are not statistically significant and the reader is likely to change their interpretation as they read these p-values (0.63, 0.49, etc.). As noted above, I think you would find the results easier to write using Spearman’s correlations for tests involving two continuous measures.

13) It’s always a risk to say that a study is the first (Line 217) and I suggest qualifying that “To our knowledge, this is the first…” or similar.

14) I think you could be stronger about the risk of ecological fallacies (Lines 239–240) than “may be liable to”. These seem all but inevitable here. This is fine as long as you keep your interpretations at the country level but I felt you might be going too far on Lines 236–237, for example. Lines 295–306 need some attention as individual-level associations would support interventions and/or public health policies, but country- and regional-level associations (Lines 300–301) will all but inevitably be subject to confounding and so cannot in themselves support such actions with special argument. Ecological associations, in my view, might support further investigations at the individual (or household) levels, but that is all they can justify unless there are particular circumstances that support a stronger interpretation, and I think you need to justify your stronger interpretation on Lines 299–306.

15) Can you be more specific about Lines 271–272?

16) You note “generally linear” associations (Line 282, also Line 46) but this cannot be directly shown with the categorisation of ECC. Linear trends with respect to ECC categories might be justified, but this is more specific than linearity. This claim would be much easier to justify from scatterplots.

16) What is the evidence for Lines 315–316 (the “modulating effect of breastfeeding”)?

17) Tables should indicate the statistical test or model used for all reported results.

18) A superscript letter system (where matching letters indicate a lack of statistically significant evidence for a difference and no letter(s) in common indicate a statistically significant pairwise difference) would make the results in Table 2 much easier to present in Table 1. I appreciate that this would mean the loss of actual p-values which contradicts my advice above, but you can always report actual p-values in the manuscript text.

Figures: I suggest thinking about and trying scatterplots for these. Note that if you retain the bar charts, you will need to think carefully whether means are appropriate and how to indicate variability. I also recommend adding the missing bar (at least a space for it) in Figure 1 to keep the layout more similar between figures.

·

Basic reporting

The article is well written, follows the guidelines, references and shows relevant results

Experimental design

The aim and scope and design are ok. Statistical analysis needs improvement: ANOVA for non-parametric data is uncommon. Kruskal Wallis would be nicer. Please check a statistician

Validity of the findings

May be if the journal can provide this it would be nice to have the full data set online.
The figures combine kg/per capita with percentage. This is highly confusing to me. Also it is common that the independent variable sugar consumption is depicted on the x-axis and the dependent variable ECC on the y-axis. Please explain this.

Additional comments

HIC sometimes countries, income level, income region is used. It would be nice to know some examples of typical LMIC countries. To me overweight is not 1 on 1 related with high sugar consumption. Total calorie consumption including fat is better correlated with overweight.
It seems that fluoride usage (toothpaste and waterfluoridation) is also a factor which varies per income of the country. Could this also be incorporated in the article for instance in the HIC sugar consumption and ECC are not related, while in the LMIC there is an association. And caries in LIC is low due to low amount of sugar consumption.

Reviewer 2 ·

Basic reporting

No comment

Experimental design

Methods section needs improvement in order to provide a more detailed description.

Validity of the findings

The discussion is well presented but should include a further aspect (the possible role of water fluoridation).

Additional comments

The manuscript presents the results of an ecological study on the associations between the prevalence of caries and overweight in preschool children, percentage of children exclusively breastfed until 6 months, and country per capita sugar consumption. The subject is relevant to public health but some aspects need to be reviewed:
Abstract
The study aim does not mention that the analysis was based on the countries’ economic classification by income level. In the last paragraph of the Introduction a different text is presented including this information, which is essential for the understanding of the study methods and conclusions.
Introduction
The following parts need corrections: Line 66 – “G Greater”; line 71 – “Also, -- individuals”.
Line 100 – The authors state that “We were however, unable to access any study that…”. I would suggest them to change this sentence, since it only indicates that they were not able to access the studies. If they do not mean that, a possible sentence would be “To our knowledge, no previous studies have…”.

Materials and methods
This section describes how each study variable was collected and analysed. An introductory part explaining the study design would be useful for a better understanding of the methodology and results. Also, how was the access to the datasets? Are they open data sources?
For the variable “Percentage of 3-5-year-old children with ECC” the source is reported, but what are the “other online databases published in the period 2007-2017”?
The variable “Per capita sugar consumption” needs a more detailed explanation. What measures where used? Does this data really refer to consumption of sugar? In some countries only data on sugar availability or acquisition is available, which does not necessarily mean consumption. More information regarding this variable is presented in the Discussion section (lines 240-241): “The data on per capita sugar consumption are national averages for all age groups; we found no age-specific data”, but should be in Methods.
The age group for the variable “Measure of overweight” is presented in two different ways: children aged 0 to 5 years old (line 135) and children less than 5 years old (lines 152 and 169). In the second version (children less than 5), those aged 5 would not be included. This should be corrected.
In “Data analysis” (lines 155-156) the information “The total number of 3- to 5-year-old children affected with ECC was calculated from a publication by El Tantawi et al [2018]” is repeated (it was presented in line 116). One-way ANOVA was also mentioned twice.

Results
I could not understand this result in lines 170-171: “The number of LICs ranged from 4-6, LMICs from 19-20, HMICs from 17-24 and HICs from 7 to 30”. What do these numbers indicate?
Table 1 – I would suggest caries and obesity instead of “sugar-related diseases” in the title. The variable “Number of 3-5 year old children with ECC in million” was not cited in Methods.
Discussion
Line 252- the use of fluoridated toothpaste was mentioned as a factor related to caries in childhood. However, another important preventive strategy is missing: fluoridation of public water supply. The same can be noted in lines 262 and 287, where access to dental care is emphasized, but there is no mention to water fluoridation. Considering the role of this population strategy for caries prevention, it should be included in this part of the manuscript. The authors should also explain why they have not included this variable in the present analysis, since it would be expected that they would influence the results. There is evidence that presence of water fluoridation may reduce inequalities in dental caries.
In 8th paragraph I would suggest the inclusion of this important paper on the role of sugars: Sheiham A, James WP. Diet and dental caries: The pivotal role of free sugars reemphasized. J Dent Res, v.94, p.1341-1347, 2015.

---

## Round 0.2 · Major Revisions

Thank you for your revisions. I feel that you have done well in improving your manuscript and most of the discussion is now careful about distinguishing between the country-level associations you have investigated and individual-level associations, although there are some places where I think the text could be made even clearer (see comments below). The two reviewers have made positive comments and there are only a few suggestions from Reviewer #2 that need to be addressed. Their point about the editing of the manuscript is important and I suggest very careful proof-reading (perhaps using an editing service from PeerJ or another source). I have some comments/queries of my own (listed below), some of which are important but I don’t imagine will require too much work on your part at this stage. I've indicated major revisions because of the number of points below and the importance of some of these rather than based on the magnitude of changes required. I look forward to seeing the next version of your manuscript.

Regarding the data, I was hoping that you would provide this as an Excel or SPSS (or some other similar format) file rather than as a Word document. The way the data is laid out in the Word document (a separate table for each outcome) would require readers to manually copy and paste this data into Excel, say, checking that they have the countries aligned correctly (there are 84, 85, 120, 133, and 142 countries for the five outcomes and countries with missing data are omitted for that outcome so this merging would not be trivial for many readers and could be error prone if they elect to do it manually) before they could replicate your analyses. Could you please provide this data in a more accessible format?

Having constructed a combined data set myself by merging the 5 outcomes using their country names, it seems to me that there is data on both ECC and sugar for 78 countries and not 77 as stated on Line 169. ECC and EBF seems both available for 57 countries and not 56 as stated on Line 170. I agree with the 53 for ECC and overweight on Line 171 but not the p-value on Line 193 (I get p=0.534 not p=0.544). Can you please check your merged data and results? Other results will change if my merged data file is in fact correct, which will be easy to determine if you provide your combined data file. I was also unable to replicate all of the numbers in your tables (overall and for each income group) so this combined data file is essential.

I’m not sure why income category is missing for 1 country with ECC data, 1 with both ECC and sugar data, and 1 with both ECC and EBF data. This country with ECC (and other) data but no income category is Argentina and the pattern of missing data (missing only overweight) is the same as 13 other countries (Austria, Belgium, etc.) which appear to be included in analyses. Is there some reason for it not being included? Does this explain the discrepancy above for the numbers of countries with usable data? The World Bank appears to have regarded them as “Upper-middle” in 2018, with them moving to “High-income” in 2019 (https://blogs.worldbank.org/opendata/new-country-classifications-income-level-2018-2019) and they are included as an upper-middle country in your data source reference (databank.worldbank.org/data/download/site-content/CLASS.xls).

Looking at the World Bank’s atlas method, this seems to be based on GNI rather than GDP as referred to on Lines 146–147 (https://en.wikipedia.org/wiki/Atlas_method). You also say “Gross National Income” yourselves on Line 144, before talking about GDP here. I couldn’t find a source for your cut points on Lines 146–147. Your reference for this data (databank.worldbank.org/data/download/site-content/CLASS.xls) includes a table note that “This table classifies all World Bank member countries (189), and all other economies with populations of more than 30,000. For operational and analytical purposes, economies are divided among income groups according to 2018 gross national income (GNI) per capita, calculated using the World Bank Atlas method. The groups are: low income, $1,025 or less; lower middle income, $1,026 - 3,995; upper middle income, $3,996 - 12,375; and high income, $12,375 or more. The effective operational cutoff for IDA eligibility is $1,175 or less.” which supports the used of GNI and provides currency cut-points that don’t match your text (please add a reference for your values), and I suggest you make the collapsing of LMIC and HMIC into MIC clear to your readers. It’s entirely possible that I’m missing something here, but if so, other readers may also need some clarification on this point.

You say that only overweight was skewed, but a histogram of sugar consumption reveals an extreme outlier (United Arab Emirates, 214.36kg) in HICs which has implications for an ANOVA here. Is this value correct/trustworthy? Note that Shapiro-Wilks provides p<0.001 for HICs (and the same overall, not that this is relevant for the ANOVAs) looking at sugar with this value left as is so this presumably was picked up in your diagnostics. Can you please recheck this point?

For my comment about currencies, please remember that there are many countries that use their own dollars and while this is often USD when not otherwise specified, this isn’t always the case ($ could refer to USD, AUD, NZD, etc.). Please provide the specific currency on Lines 146–147.

For my comment about descriptive statistics, please note that the IQR is a single number (see https://en.wikipedia.org/wiki/Interquartile_range). In Table 1 you appear to be giving the 25th and 75th percentiles instead. This is fine, but if you are giving the percentiles you should label them as such; and if you mean to give IQRs, these are the differences between these two percentiles.

Regarding stratification, this doesn’t allow statements about effect modification (this requires a direct comparison of the groups, which is easiest when all groups are analysed together). It is possible, for example, for two groups to have correlations in the same direction but for there to be statistical evidence for effect modification (i.e., a statistically significant interaction or difference in correlations) and for two groups to have correlations in opposite directions and for there to be no evidence for effect modification. The reference you provided for this produces a webpage that does not seem to be publically accessible, but searching for the course found the overview page http://sphweb.bumc.bu.edu/otlt/MPH-Modules/BS/BS704_Multivariable/ which notes, correctly, that “Effect modification occurs when the magnitude of the effect of the primary exposure on an outcome (i.e., the association) differs depending on the level of a third variable. For example, suppose a clinical trial is conducted and the drug is shown to result in a statistically significant reduction in total cholesterol. However, suppose that with closer scrutiny of the data, the investigators find that the drug is only effective in subjects with a specific genetic marker and that there is no effect in persons who do not possess the marker. This is an example of effect modification or ‘interaction’. The effect of the treatment is different depending on the presence or absence of the genetic marker. Multivariable methods can also be used to assess effect modification.” As they point out, this involves testing for evidence of interactions and this requires analyses using the full data set. While Pearson’s correlations can be directly compared (using Fisher’s z-transformation), comparing Spearman’s correlations requires bootstrapping (at least I am unaware of alternatives). You can highlight numerical differences between strata and not look at these comparisons statistically, but in that case, it is important to be clear to the reader that this is then not testing for effect modification (e.g. Line 32 and elsewhere).

Line 28: Perhaps either “children with overweight status” or “children classified as overweight” here? See also Lines 163, 171, and perhaps elsewhere.

Line 35: I think you need to make it clear to the reader here or around here in the abstract that only countries with ECC data are included and how many of these there are in each income category.

Line 42: Perhaps “modest” rather than “moderate” but note that Pearson’s correlations of 0.5 are regarded as large (see https://en.wikipedia.org/wiki/Effect_size#Pearson_r_or_correlation_coefficient for details and references) and I would personally use the same interpretation for a Spearman’s correlation of this magnitude.

Line 58: “carbohydrateS”. See also Line 69 and perhaps elsewhere.

Line 91: Perhaps “…no study HAS addressed…”

Line 94: I suggest deleting “significantly” from “A significantly large number of studies” or replacing “significantly large” with another adjective.

Lines 96–98: This sentence (“Yet, evidence on macro-level impact of risk factors for diseases give insights about structural interventions that may help in achieving population-level disease control.”) needs some rewording.

Line 111: Please reference these sources here and/or indicate that the sources will be described below.

Lines 115–117: Please add the references here so the reader can access these directly. You could also start with the reference to El Tantawi, et al. (2018) (here, rather than on Lines 120–121) if this is exactly what was done here and then briefly explain what was done there to obtain this information (preferably with references also included here). At the moment, a reader will be left at least momentarily confused about what was done on this lines.

Line 125: I’m not entirely sure why there is no year for this reference. The document properties for the link you provide give a creation date of “May 26, 2018” (File->Document details), which seems appropriate to use here along with the date of access that you also provide.

Line 126: “THESE data…” (data is plural, as you indicate on Lines 118, 124, etc.) See also Lines 169, 172, and perhaps elsewhere.

Lines 130–131: The “author” list varies between here and the reference on Lines 454–457 (I am assuming these are the same source.)

Lines 131–135: A reference would be useful here.

Lines 153–154: It’s not clear here whether these tests were performed overall (which would inform summary statistics for the entire sample), by income group (which would inform summary statistics for the stratified results and statistical tests comparing these groups), or in both ways.

Line 158: You need to specify which post-hoc test was used following Kruskal-Wallis tests, including any adjustment for multiplicity.

Lines 161–162: Rather than “with exclusive breastfeeding”, perhaps “who were exclusively breastfed” or similar? See also Lines 170, 190, 196, 202, and possibly elsewhere.

Lines 172–173: I wonder if giving the minimum–maximum numbers of countries would be useful to readers here.

Lines 180–182: Because you are using a non-parametric test for overweight status, it is more usual, but not required, to report medians rather than means. Can you either change this or argue in favour of continuing to present means?

Lines 185–187: Assuming this is talking about absolute numbers, this isn’t a useful statement as it depends as much on the relative populations of the three income groups as it does on the relative proportions with ECCs. Your sample of countries isn’t random or necessarily representative, so even with that information, it would be slightly limited in interpretation. If you feel you can justify this point, that’s fine, but without a strong justification, it should be deleted.

Line 188: I think you can delete “the direction and magnitude of” as these are what a correlation is.

Line 189: “modified” is word suggesting a causal claim and so cannot be justified here. In any case, this would require either formal statistical testing for effect modification or making it clear that the correlations were numerically different between income groups. The latter of these is inevitable from a numerical perspective and without these p-values, or for tests at approximately the 0.01 level, 95% CIs around the correlations, identifying potentially interesting differences between groups would have to be argued for based on the differences/ratios themselves. If you are able to incorporate bootstrapping to test for differences, or some other approach to this, it would greatly enhance your ability to draw conclusions here.

Lines 191–195: Several of these correlations are not statistically significant and this needs to be made clearer in the wording around the correlations (as you do on Line 200, for example), not just by the p-values. Or you could delete this text as you then go over the statistically significant correlations on Lines 195–198.

Line 209: This describes what appears to be a statistically non-significant correlation between ECCs and sugar in LICs as if it were statistically significant, and I think the result for MICs could also be made clearer. Perhaps you could instead refer to the figures back around Line 188 (and delete Lines 206–218) before you talk about the correlations, using the figures to prepare the reader to understand the associations visually, so you are not going over this material a second time here?

Lines 226–227: Again, unless you change the analysis approach, you will have to approach this point in a different way as you do not have statistical evidence that these associations differ. Similar point for Lines 233–234.

Lines 237–238: As mentioned previously, you are looking at country-level data and so cannot make interpretations for individual-level associations with any degree of confidence. You need to make it clear to the reader that you mean country-level associations here.

Line 238: I’d hyphenate “country-specific” here and elsewhere.

Lines 239–240: This sentence appears to simply generalise the final point from the previous sentence and could be replaced by “along with other risk factors” being added to that previous sentence.

Lines 242–244: I don’t see how you have established this as a “probability” here. The association between ECC and sugar was found only in HICs and there was much less data available on weight status (n=7 versus n=30 from my data set) so that particular test had much less power (the values of Rho you report in Table 2 are 0.51 and 0.40 and so are not particularly different in the numerical sense). I think that you might be over-interpreting non-statistically significant results here and if you were to compare the strengths of those two correlations through bootstrapping, I would not expect you to find any evidence of a difference, despite one being statistically significant and one not.

Lines 250–255: I would interpret the overall correlation disappearing when you looked at income groups as suggesting that there is at least one third factor (either wealth or something associated with it) that confounds this association. I don’t think that this raises new queries (Line 255) regarding individual-level associations as you cannot make statements about individual-level associations with a meaningful degree of confidence here.

Line 258: Again, rather than ecological fallacies being possible, they are for all intents and purposes inevitable here. You just cannot say anything about individual-level associations here with confidence (it all has to be speculation) as your analyses are based on country-level data.

Lines 258–260: This sentence reads awkwardly for me. Please try to make it clearer.

Line 261: I think you could delete “details of” here without loss of meaning.

Line 270: Needs rewording “The need…is urgently needed…”.

Line 298: The just mentioned studies would need to be country-specific and look at individual-level data to inform interventions at this level and I think this could be clarified for the reader (e.g. Line 295 “Individual-level studies are needed…”).

Line 323: Perhaps “…may AFFECT their risk for ECC…” (or reduce or decrease).

Line 324: Perhaps “observations” or “possibilities” rather than “suggestions”.

Table 1: Please indicate the sample sizes, which will be for cells rather than for columns or rows given the data availability. I appreciate that this is messy, but readers will need to see this information one way or another. Note that the SD for sugar consumption is high relative to the mean for HICs, indicating the positive skewed outliner. Note as already mentioned that IQR is a single number (https://en.wikipedia.org/wiki/Interquartile_range) and not the two percentiles shown here. You can show the 25th and 75th percentiles instead of the IQR (which is their difference) but the table needs to label them as such. As commented on above, the number of children (last row) depends on both the population and prevalence and so I’m not sure that it’s useful here without those two details also presented (or you could delete it), unless you can make an argument otherwise? For overweight, I wouldn’t show both the mean and the median, and the median is more usual for non-parametric tests. The post-hoc test following Kruskal-Wallis needs to be included in the notes (as you for the post-hoc tests following ANOVAs).

Table 2: I think readers will again want to see sample sizes, which will be messy but I don’t think there is a way of doing this except cell-by-cell here.

Figures 1–3: I think something has gone wrong with the data as you appear to have too many MICs and many of these have zero for ECCs (the points along the x-axis). Please ensure that both the x-axis and (especially the) y-axis are labelled in each subfigure.

·

Basic reporting

The authors have adressed the remarks of the reviewers appropriately.
The main results and conclusions can now be accepted for publication.

Experimental design

no comment

Validity of the findings

no comment

Additional comments

Dear Authors,
Thank you for adressing all the issues that the reviewers have mentioned.
The manuscript is now much better.
For a dentist however the perspective of population with a disease seems to loose contact with the individual patient.
However, the population based perspective helps understand group/ population dynamics trends.
Thank you for this perspective.
Based on the recent trend of national initiated population and/ or industry regulated restrictions in sugar usage monitoring of populations trends will become more interesting.
The coming years the effects of such regulations need to be monitored and evaluated.

Reviewer 2 ·

Basic reporting

The English language in the manuscript requires review, specially the new sentences included in the text.

Experimental design

No comment.

Validity of the findings

Part of the conclusion is not linked to the original research question. Please see further comments on this aspect in "General comments for the author".

Additional comments

The authors have adequately addressed most of my previous concerns.
The aspects that need further review are in the Abstract:
In the Aim, overweight was considered as “an indicator of high sugar consumption”. Although extrinsic sugars may be related to overweight and obesity, there is no clear evidence that they are indicators of these conditions. I would therefore suggest the exclusion of this term. The new sentence would be “This ecological study determined the association between the prevalence of early childhood caries (ECC), overweight and country per capita sugar consumption”.
In the Conclusion, the authors stated that “…there was no clear relationship between the two diseases suggesting that planning for their control using a common approach may not produce similar impact”. The possible implication (“suggesting that planning for their control using a common approach may not produce similar impact”) should be removed, since it is not supported by the study methodology and findings. It is not in the Discussion.

---

## Round 0.3 · Minor Revisions

Thank you for your careful revisions and thoughtful responses. You have done an excellent job at revising your manuscript in light of the reviewers’ and my comments and the comments from reviewers are now fully addressed. There are still a few remaining queries from my comments that require attention, but my hope is that these will not present you with too much difficulty and that, subject to appropriate responses and no new issues emerging, I will be able to quickly accept the next version of your manuscript.

The treatment of the sugar data from the United Arab Emirates seems to be the one substantial issue remaining. This very high value, 214.36kg in the previous version, appears to be a correct value as far as I can tell (Googling “sugar intake United Arab Emirates” finds a variety of supporting news articles, etc.). In your response, you say that “We removed the outlier (the UAE value) and the histogram plus normality test improved so we kept it removed. Thus, only overweight status was not normally distributed.” However, unusual but real data cannot be removed simply because it is unusual. This is a situation where a log-transformation could be considered for sugar intake (“pulling” in the very large value), or non- or semi-parametric approaches considered (such as Kruskal Wallis as you already use, but quantile regression would be an option for the regression analyses). An alternative approach would be to include the UAE values in the primary analyses (as it seems to be a real value) and to also check results with it removed (as it is so unusual). Note that there are other values in the dataset that are almost as extreme (Mauritania 86.69, Bahrain 93.03, Belize 105.70, and particularly Djibouti 193.32, although ECCs data is not available for any of these) suggesting that per capita sugar consumption simply has a very long tail to the right with UAE at the extreme. I wonder if one of log-transforming sugar intake or running models with (first) and without (as checks for the robustness of the results) UAE sugar data would be the best approach here and would be interested in your thoughts. If you take the second of these options, you would present one set of results (using all data makes the most sense since the data is all valid) and then note whether or not anything changed in an important way following exclusion of an unusual data point (UAE’s sugar data) without needing to provide a second set of tables for those analyses (although you could do this as a supplement if you wished).

Related to this, there still needs to be some discussion of model diagnostics used in the statistical methods for the regression models (Lines 150–155), which might include looking at the normality of residuals, their homoscedasticity (equal variance), and the linearity of the association through histograms, formal statistical tests, and/or scatterplots. Remember that linear regression doesn’t make assumptions about the data itself beyond it being continuous and independent, all of the other assumptions concern the model residuals.

Note that while Mann-Whitney U seems a “logical” post-hoc test following Kruskal-Wallis, a preferred post-hoc test is Dunn’s test. A brief description of this can be found on https://stats.stackexchange.com/tags/dunn-test/info and note that SPSS implements this as a post-hoc test.

Finally, the text on Lines 196–202 all concerns non-statistically significant results except for the MIC exclusive BF result. While it would be fine to describe the pattern of directions in these associations, it needs to be made very clear to the reader that there is no evidence supporting these associations from your analyses before describing them. Note that the word “associated” is often read as indicating statistical significance. You also need to be careful in the Discussion. For example, Lines 217–218 talk about a “direct association” first (without making it clear what a direct versus indirect association would mean, see also Lines 189 and 196) and only then mentions the lack of statistical significance. The same applies to Lines 218–221. Lines 230–231 neglect to indicate statistical significance. As this is a hypothesis generating study, it’s fine to point out patterns of associations despite p>0.05, but this needs to be done carefully and by ensuring the reader appreciates the statistical significance or lack thereof first. I’ll ask you to carefully check your results and discussion sections to make sure that readers will immediately appreciate when you are reporting a statistically significant versus a non-statistically significant finding.

You have done a good job of communicating the macro level of the analyses in most places, although Line 237 seems to me to go too far in describing this association as a “risk factor”, which suggests an individual-level association to me and would read like a causal claim from observational data.

There are some specific comments below that should be addressed also:

Line 2: You’re missing a comma between “overweight” and “sugar consumption”.

Line 3: Same for between “low” and “middle”.

Line 42: I’d delete the “A” in “A significant effect modification…”, so starting the sentence “Significant effect modification…”

Line 43: Do you mean “categories” rather than “countries” here?

Line 45: I was surprised that the one statistically significant association (ECCs and exclusive BF in MICs wasn’t mentioned here). The sentence on Lines 41–42 (“No significant differences were observed in the prevalence of ECC between LICs and MICs (P= 0.961) or between LICs and HICs (P= 0.133).”) would seem to be a candidate for removing if you needed additional words to add this in here.

Line 145: Either “by Scheffe’s test” or “by a Scheffe test” here.

Line 145: “comparisonS.” (adding “s”)

Line 146: “THE Kruskal Wallis test…” (adding “The”)

Line 147: “comparisonS.” (adding “s”)

Lines 147–148: If you wish to argue for continuing to use Mann-Whitney U, this could be “using THE Mann Whitney U test”. The phrase “adjustment for multiple testing” doesn’t tell the reader what adjustment was made and this detail needs to be added here, irrespective of whether Mann-Whitney U or Dunn’s test is used.

Line 151: “associationS” (adding “s”)

Line 154: “associationS” (adding “s”)

Line 171: The reader might appreciate being reminded about the units for the sugar consumption measure here (e.g. 8.11 kg).

Lines 173–179: Do you feel that the second decimal place is warranted for these percentages, or is this too precise?

Lines 186–187: It might be useful to indicate the direction of this association to the reader here, to save them needing to check the table for this information.

Line 235: “…it was noT possible…”

Line 242: While “modifying” is correct in its usual English sense of changing, given your references to effect modification, perhaps delete “modifying” here or replace with “mediating”.

---

## Round 0.4 · Minor Revisions

Thank you very much for your revisions and your patience with the reviewing process during this challenging period.

While I would very much like to be able to accept your manuscript, there are still more language issues than I think is advisable to try to correct in proofing (a few is fine but the list of those that I noted below is a little longer than that). Please do proof-read the manuscript carefully for any that I might have missed. You are of course welcome to use an editing service (for example, PeerJ’s) if you would like the reassurance that all such issues should have been addressed.

There are still some instances of what seem to me to be initial, and sometimes only momentary, potential over-interpretations of non-statistically significant findings. See my comment starting “Line 198” below. My previous comment “make sure that readers will immediately appreciate when you are reporting a statistically significant versus a non-statistically significant finding” wasn’t perhaps as clear as it could have been and I think you need to be careful to try not to describe associations and then only afterwards note that some of these were not statistically significant. As an example, on Lines 229–230, you say “Per capita sugar consumption was directly associated with the prevalence of ECC in MICs and HICs, though the association was only significant in MICs.” A reordering edit to “While only statistically significant for MICs, per capita sugar consumption was directly associated with the prevalence of ECC in MICs and HICs.” means that the reader will never interpret “per capita sugar consumption was directly associated with the prevalence of ECC in MICs and HICs” as indicating statistical significance, which they could momentarily do with the current wording.

I’d also like to see some (brief) comment(s) added to the manuscript about the UAE sugar consumption outlier so the reader is aware of this unusual data point (otherwise, I don’t think they would necessarily realise this without them looking at the raw data or the regression diagnostics supplement). This point doesn’t need much, just an acknowledgement in the text of the outlier (perhaps early in the results) and a mention of how the results would change if it was removed (at least in the discussion). The options I suggested last time were log-transforming sugar consumption, quantile regression to model medians, and “to include the UAE values in the primary analyses (as it seems to be a real value) and to also check results with it removed (as it is so unusual).” with a further clarification that “you would present one set of results (using all data makes the most sense since the data is all valid) and then note whether or not anything changed in an important way following exclusion of an unusual data point (UAE’s sugar data) without needing to provide a second set of tables for those analyses (although you could do this as a supplement if you wished)”. While you have apparently chosen this last option, I couldn’t find any evidence of the second half of this recommendation. Can you please either justify not doing this or add appropriate reference(s) to the manuscript about UAE’s unusually high consumption? You do this quite nicely in the supplement by showing the scatter plot with and without UAE.

I don’t think your manuscript is far at all from being able to be accepted and I will hope that you are able to address the points above and below, which you are welcome to reject with justification, without too much difficulty.

Line 36: “The percentage of infants who WERE exclusively breastfed…”

Line 40: Should this be “…significant modifier of the associations between the prevalence of ECC AND per capita sugar…” so the two p-values can be interpreted?

Lines 41 and 47: For some reason these two p-values are given to two decimal places only. If there is a trailing zero, this should be shown so the reader doesn’t wonder if a digit is missing. I did wonder if you were using a rule based on the magnitude of the p-value itself, but you give 0.021 to 3 decimal places on Line 38 and then 0.03 and 0.02 to only 2 (Lines 41 and 47 respectively). See also Lines 179 (0.01), 192 (0.02), 210 (0.02), 217 (0.26), Table 1 (0.03), Table 2 (0.03 and 0.26), and perhaps elsewhere. Please check the manuscript thoroughly to ensure that the presentation of p-values is consistent throughout.

Line 46: Should this be “…between the prevalence of ECC AND the percentage of children…”?

Line 78: I think you wanted a comma in “Multiple pathways have been suggested for this relationship, one of which is…”

Line 112: I think you might mean “…multiplied BY 100.” here.

Line 117: Do you mean “…measured in per capita kilogramS FOR EACH country for 2017”?

Line 122: UNICEF is mentioned twice in the list here.

Line 127: I think you have an extra “children” (the first one) in “…percentage of children 0-5-year-old children who met the definition.”

Line 131: “was” here should be “were” due to the plural “estimates”.

Line 146: I think you want “Kolmogorov-Smirnov AND Shapiro Wilk tests,…” (as the tests refers to two things and note “Wilk” and not “Wilks” as is currently in the manuscript).

Line 159: You want a plural in “…were checked to verify models’ assumptionS.”

Line 162: “variance” is singular, so you want “was” rather than “were” in “…variance of residuals WAS constant.”

Line 163: If you have a comma after “size” on Line 164, you’ll want one before “as” here to set off this clause.

Line 168: I think “per income region” can be deleted from here.

Line 178: Should this be “and ECC AND the three income regions.” as you’re talking about the association between each of these variables and income region?

Line 182: I think this should be “(mean= 15.41 vs 40.01, P< 0.001).” as you’re talking about HICs being lower than the other two groups.

Line 198: This supplements and expands on my comment above. This coefficient (“regression coefficient= 0.18”) does not appear to be statistically significant in Table 2 despite the description here implying that it is. Only the result for MICs (Line 198) is statistically significant in Table 2 and that needs to be made clear in the text around here (Line 197–205). Please check that non-statistically significant results are clearly indicated as such whenever they are mentioned. It’s fine to describe directions and to note that a coefficient is higher or lower than another, but I suggest making the point that the associations are not statistically significant clearly and before presenting the coefficients/association to the reader so they can quickly and linearly interpret the results. As you give p-values for earlier results, you might find including additional p-values here useful for indicating statistical significance to readers. Similar issues arise in the paragraph on Lines 207–214, although you are clearer there about the only statistically significant result being for MICs (Lines 209–210). The sentence on Line 229–230 doesn’t address the non-significance of the results outside of MICs until the second half, which will cause the reader to potentially revise their interpretation of the first half. As suggested above, noting non-significance before describing the results will save the reader from the risk of reinterpreting statements like this. The same point would apply to Lines 232–233 and Lines 244–245. I’ll ask you to check the manuscript thoroughly to ensure that whenever a non-statistically significant result (i.e., all associations from Table 2 aside for sugar and breastfeeding in MICs) is presented or discussed, readers will not potentially need to change their interpretation after the association is described. I also note that Lines 289–291 would, if read in isolation, suggest a statistically significant association in HICs to me.

Lines 199 and 200: I think the two instances of “high” here should be “higher” as you’re describing slopes rather than levels of these variables (they might be relatively high without being absolutely high). Similarly “high” and “low” on Line 202.

Line 214: I’m not sure why you have so many decimal places here (“partial eta squared= 0.000002”), particularly as you use an inequality back on Line 205 with fewer decimal places (“<0.0001”). Given the other figure on that line (“0.003”), I suggest you use 3 decimal places for all eta-squared values and <0.001 for very small values but I’d be happy to hear reasons for other approaches.

Lines 223–224: I’m not sure about this sentence (“The results highlight a complex relationship between these factors and the prevalence of sugar-related diseases (ECC and overweight).”) Both ECC and overweight are included in the just mentioned factors and so as written the sentence seems to be referring to them twice. Is something missing from the second part of the sentence or am I misreading something here?

Line 244: Perhaps “THE [d]irect association…” here.

Line 344: This line should be indented?

Line 440: This reference seems incomplete. Is this a report, webpage, or something else?

Line 447: Minor typo in “United Nations Children’s Fud nd”

---

## Round 0.5 · accepted · Accept

Thank you for your patience and your revisions, which have allowed me to accept this version of your manuscript. Congratulations!

I have noted some minor typos below for you to address during the proofing process. Suggested edits are indicated in capital letters where appropriate.

Line 22: “…examined the associationS between…”

Line 44: Perhaps either “…was significant ONLY in MICs (P= 0.002),…” or “…was significant in MICs (P= 0.002) ONLY,…”.

Line 75: Need a comma as indicated in “…suggested for this relationship, one of which…”

Line 90: “…on the relationshipS between…”

Line 95: “…the relationshipS between…”

Line 109: “…examined in the country, multiplied BY 100.”

Line 119: Should this be “…UNICEF, AND the World Bank [2018] and UNICEF [2018],…”

Line 128: “…life WERE obtained from the WHO Global Health…”

Line 136: Note there is a missing space in “-GNI” (c.f. Lines 136 and 137).

Line 143: I suggest replacing the comma with “and” here, i.e. “…Smirnov AND Shapiro Wilks tests…”

Line 145: Note there is an accent in “Scheffé’s”.

Line 156: “…verify models’ assumptionS.”

Line 160: If you want to keep the comma after “size” on Line 161, you need a matching one here: “…partial eta squared, as a measure of effect…” Or you could instead delete the comma on Line 161.

Line 174: “…the associationS between…”

Line 175: “…and ECC AND the three income regions…” (the associations are between the initially listed variables and income regions here).

Line 187: For consistency “MICs - middle-income countries” should be “MICs: middle-income countries” (c.f. LICs and HICs just before and after). See also the version of Table 2 at the end of the manuscript, which differs from the version inline here.

Line 189: Based on Lines 146–149, this should read: “Kruskal Wallis test was used followed by DUNN-BONFERONI post-hoc test for pairwise comparisonS…” or the text in Table 1 at the end of the manuscript. (These two versions of the table were not identical for this text.)

Line 190: Table 1 notes “Scheff test” which should be “Scheffé test” (note the spelling of the name, which is correct on Line 145, including the accent). And “…pairwise comparisonS…”

Lines 197 and 192: I think the phrase “direct relationship” might be clearer as “POSITIVE relationship” in each of these locations, although you might prefer “POSITIVE ASSOCIATION” for consistency with the following text.

Line 201, 217, 240, 254: I think the phrase “direct association” might be clearer as “POSITIVE association” in each of these locations. On Line 254, “A” should be added at the start of the sentence.

Line 235: I think you could delete “First,” here (there is no “second” or “third” in the text).

Line 236: Perhaps “…income regions, AND HICs had the…” rather than the colon (“:”) here.

Line 239: As with the comment for Lines 201, 217, 240, and 254, I think “was POSITIVELY associated with…” might be clearer.

Line 266: You’ll need to edit this as “so” and “therefore” are saying the same thing, perhaps by deleting “therefore, ” from “…and therefore, so it was not possible…”

Lines 322–324: This sentence (“However, the Institute provides estimates for diet high in sugar-sweetened beverages, which limits the potential to quantify the prevalence, incidence, burden and mortality of diseases attributed to consumption of all sugars [Institute of Health Metrics, 2017].”) needs some attention for clarity.